# Examination of alternative eGFR definitions on the performance of deep learning models for detection of chronic kidney disease from fundus photographs

**Songyang An[1,2]\*, Ehsan Vaghefi[1,2], Song Yang[2], Li Xie[2], David Squirrell[2,3]**

**1** School of Optometry and Vision Science, The University of Auckland, Auckland, New Zealand, **2** Toku Eyes Limited NZ, Auckland, New Zealand, **3** Auckland District Health Board, Auckland, New Zealand

\* Songyang.an@auckland.ac.nz

**Data Availability Statement:** Unfortunately, as we chose to use the UK Biobank dataset for our study, there are restrictions on data sharing. Under clause

## Abstract

Deep learning (DL) models have shown promise in detecting chronic kidney disease (CKD) from fundus photographs. However, previous studies have utilized a serum creatinine-only estimated glomerular rate (eGFR) equation to measure kidney function despite the development of more up-to-date methods. In this study, we developed two sets of DL models using fundus images from the UK Biobank to ascertain the effects of using a creatinine and cystatin-C eGFR equation over the baseline creatinine-only eGFR equation on fundus image-based DL CKD predictors. Our results show that a creatinine and cystatin-C eGFR significantly improved classification performance over the baseline creatinine-only eGFR when the models were evaluated conventionally. However, these differences were no longer significant when the models were assessed on clinical labels based on ICD10. Furthermore, we also observed variations in model performance and systemic condition incidence between our study and the ones conducted previously. We hypothesize that limitations in existing eGFR equations and the paucity of retinal features uniquely indicative of CKD may contribute to these inconsistencies. These findings emphasize the need for developing more transparent models to facilitate a better understanding of the mechanisms underpinning the ability of DL models to detect CKD from fundus images.

## Introduction

Chronic kidney disease (CKD) is a systemic condition characterized by the progressive deterioration of kidney function over time and a corresponding decrease in the glomerular filtration rate (GFR) [1]. CKD risk factors include obesity, diabetes, hypertension, and nephrotoxin exposure. Although GFR can be measured by methods such as the urinary clearance of inulin [2], the inherent difficulties in this procedure mean that in practice, clinicians rely on inferential methods which use surrogate markers [3] such as estimated glomerular filtration rate (eGFR), in-urine albumin-creatinine ratio, and the underlying physiological context of the symptoms [4] to assess underlying renal functionality.

2.2 of the UK Biobank material transfer agreement (https://www.ukbiobank.ac.uk/media/p3zffurf/biobank-mta.pdf), "the applicant shall not share, sub-license, disclose, transfer, sell, gift or supply the materials to any other unauthorized third party." However, as per clause 2.5 in the MTA, this link (https://www.ukbiobank.ac.uk/enable-your-research/apply-for-access) provides the procedures needed to request the data. The institutional point of contact for accessing the data is access@ukbiobank.ac.uk. Once UK Biobank has granted permission, the minimum dataset can be accessed from https://huggingface.co/san727-UOA/PONE-D-23-21560 upon reasonable request.

**Funding:** S. An was awarded a R&D Fellowship Grant, TEYES2101/PROP-81260-FELLOW-TEYES by Callaghan Innovation, https://www.callaghaninnovation.govt.nz. The sponsor or funding organization had no role in the design or conduct of this research.

**Competing interests:** I have read the journal's policy and the authors of this manuscript have the following competing interests: E. Vaghefi is a co-founder and CEO of Toku Eyes Limited New Zealand. D. Squirrell is a co-founder and medical advisor at Toku Eyes Limited NZ. S. An, S. Yang and L. Xie are employees of Toku Eyes Limited NZ. The authors report no other conflicts of interest in this work.

The recent successes of artificial intelligence (AI) and deep learning (DL) have encouraged the development of alternative methods to screen for CKD. In light of the proposed linkage between the retina and the kidneys [5–7], along with the plethora of useful biological information researchers have extracted from the fundus using DL methods [8–10], research groups have applied DL to either directly diagnose CKD from fundus photographs [11–15], predict physiological markers from fundus photographs [16], or to create synthetic markers [17] indicative of CKD from fundus images.

To date, all publications focusing on the direct diagnosis of CKD from fundus photographs [11, 12, 15, 18] have used convolutional neural networks (CNN) as the DL architecture of choice. Furthermore, all groups have elected to use eGFR equations, such as CKD-EPI 2009 [19] or MDRD [20], based on serum creatinine, age, gender, and race as ground truth for renal function. One limitation of this approach lies in serum creatinine's sensitivity to factors unrelated to kidney function. As serum creatinine is a waste product of muscular degeneration, factors such as vigorous exercise, chronic glucocorticoid therapy, and hyperthyroidism could also alter its levels in the bloodstream [21]. To remediate this shortcoming, the research group that created the serum creatinine eGFR equations [19, 20] has developed more recent versions to include serum cystatin C alongside serum creatinine [19, 22, 23]. It has been proposed that serum cystatin C is a more discriminative biological marker when compared to serum creatinine [24–26], and the updated equations encompassing both biomarkers are more reliable [22, 27].

This study aims to quantify the influence of these two definitions of eGFRs—creatinine only and creatinine plus cystatin C on the convolutional neural network's ability to classify CKD from fundus photographs. To this aim, we developed and compared an ensemble of models on the UK Biobank dataset for each eGFR definition. Standard metrics, including ROC AUC scores, sensitivity, specificity, and F1 scores, were used to measure the performance of the models.

## Methods

### Study participants

The UK Biobank is an open-access research resource containing health information for over half a million participants from the UK that were initially recruited from 2006–2010, with follow-up visits occurring until 2022. During the initial assessment and the first repeated assessment visits, 175,788 non-mydriatic, 45° primary field of view, macula-centered fundus photographs from both the left and right eyes were captured from 85,707 individuals using the TOPCON 3D OCT 1000 Mk 2.

### Ethics statement

This study is a retrospective study of the medical records captured in the UK Biobank. UK Biobank has been granted approval from the North West Multi-Centre Research Ethics Committee as a Research Tissue Bank approval (RTB). This approval waives the requirement for informed consent, which means researchers with applications approved by the UK Biobank do not require separate ethical clearance. The RTB approval was granted in 2011 and was last renewed in 2021. As per UK Biobank's de-identification protocol, the UK Biobank provides de-identified data to researchers in a manner which preserves the anonymity of its participants, and as far as practically possible, does not enable participants to be inadvertently identified. A material transfer agreement between our research group and UK Biobank was finalized on the 28[th] of March 2022 under application number 86299.

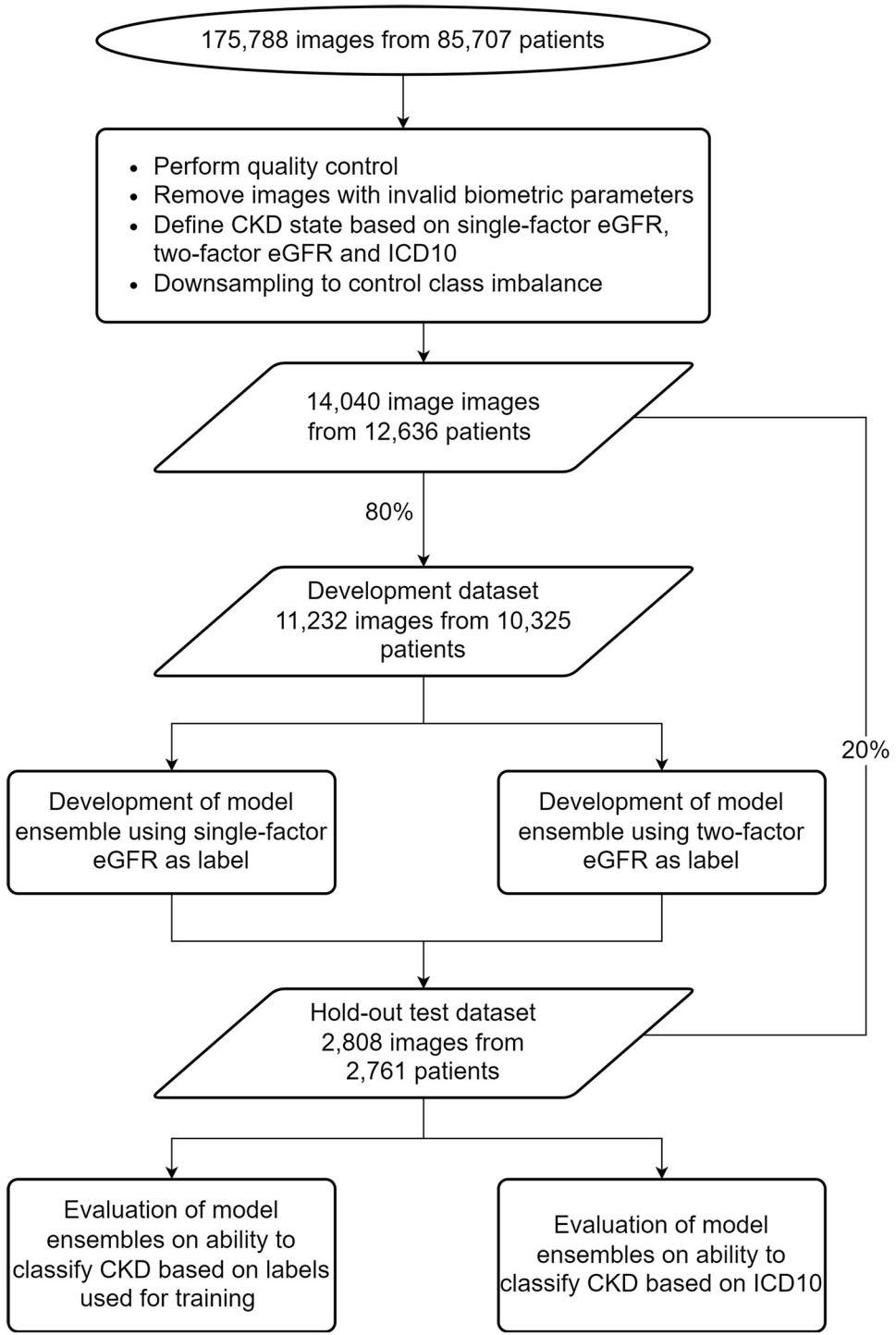

**Fig 1. Flow chart illustrating experimental methodology.** Flow chart illustrating experimental methodology.

## Experiment overview

The methodology used in this study is illustrated in Fig 1. A summary of the experimental method is as follows:

1. A training and hold-out testing dataset was created from the collection of fundus images available from the UK Biobank.

2. Three methodologies were used to define binary patient groups for CKD disease status:

   a. Using a single-factor (serum creatinine-only) eGFR.

   b. Using a two-factor (serum creatinine and serum cystatin-C) eGFR.

   c. Using clinical diagnosis based on ICD 10 codes for CKD.

3. Two ensembles of standard DL models for classifying CKD from fundus images were trained on the two eGFR definitions.

4. Using the hold-out testing dataset, the two model ensembles were tested for their ability to predict CKD disease status according to two definitions:

   a. CKD as defined by the eGFR method used to develop the model ensemble.

   b. CKD as defined by ICD10 codes.

5. ROC AUC, sensitivity, specificity, and F1 were used to measure the models' performances. Paired t-tests were used to determine the statistical significance of the results.

All deep learning models were developed using the Pytorch library and trained on a workstation with AMD Threadripper 3970X CPU, 2 RTX A6000 GPUs connected via NVLink, and 256GB of RAM.

## Dataset description

175,788 fundus images from 85,707 individuals were obtained from the UK Biobank. A DL image screening system similar to the one used in our previous study [28] was created to screen for poor-quality images. In brief, a random set of images were first sampled from the UK Biobank dataset. Then based on the New Zealand Diabetic retinal screening, grading, monitoring, and referral guidance [29], images were manually sorted into good quality, poor, over-exposure, under-exposure, excessive occlusion, and non-fundus (eyeball). A DL multi-class classification model was trained on this set of images and then executed over the entire collection of images.

For those individuals who had good-quality images, key biometric parameters, including serum creatinine, cystatin c, age, sex, ethnicity, and date N18 (chronic renal failure) first reported (UK Biobank field 132032), were retrieved. Where multiple parameter values were recorded over multiple sequential visits, they were matched to the corresponding fundus photographs taken during the same visit. The CKD-EPI 2021 [22] serum creatinine (single-factor) and serum creatinine plus cystatin C (two-factor) eGFR equations were then used to estimate kidney function for all individuals.

Down sampling was then implemented to address the potential issues with class imbalance [30]. Individuals with either of the two eGFR definitions below 60 mL/min/1.73m$^2$ were identified. All remaining individuals were then randomly down sampled to achieve a 1:8 ratio between the former and latter groups. A 1:8 ratio was chosen because existing studies showed models could generalize given a case-to-control ratio of 1:4 to 1:16 [11, 18].

An 80:20 split was carried out to generate the development and hold-out test datasets. The development dataset was then further subdivided using an 80:20 ratio into the training and validation subsets. The training subset was used to train the model, and the validation subset

**Table 1. Dataset summary table.**

| | Overall | Development (80%) | Hold-out test (20%) |
|---|---|---|---|
| **Number of Images** | 14,040 | 11,232 | 2,808 |
| **Number of Participants** | 12,589 | 10,326 | 2,750 |
| **Individuals with eGFR < 60 mL/min/1.73m²** | 942 | 854 | 285 |
| **Corresponding number of images for Individuals with eGFR < 60 mL/min/1.73m²** | 1,560 | 1,249 | 311 |
| **Male (%)** | 46 | 46.2 | 45.2 |
| **Age** | 57.2 (8.36) | 57.3 (8.34) | 56.9 (8.43) |
| **BMI** | 27.3 (4.8) | 27.3 (4.79) | 27.4 (4.86) |
| **eGFR (mL/min/1.73m2)** | 91.6 (18.2) | 91.6 (18.1) | 91.7 (18.4) |
| **hBA1C (mmol/mol)** | 36 (6.32) | 36 (6.3) | 36 (6.39) |
| **Cholesterol (mmol/L)** | 5.65 (1.14) | 5.66 (1.14) | 5.62 (1.13) |
| **HDL cholesterol (mmol/L)** | 1.48 (0.392) | 1.48 (0.393) | 1.47 (0.386) |
| **Mean systolic blood pressure (mmHg)** | 137 (18.4) | 137 (18.4) | 137 (18.2) |
| **Mean diastolic blood pressure (mmHg)** | 81.3 (10) | 81.4 (9.99) | 81.2 (10.1) |
| **Diabetes (%)** | 5.52% | 5.36% | 6.16% |

Summary of data used for study. 14,040 images were used in total, with 11,232 allocated to the development set and 2,808 hold-out test set. Individuals with eGFR < 60 mL/min/1.73m² was the total number of individuals with either serum creatinine only eGFR or serum creatine and cystatin C eGFR < 60 mL/min/1.73m².

was used for early stopping to prevent overfitting. A summary of the dataset is given in Table 1.

## Model ensemble construction

Five models were constructed for each model ensemble:

1. Fundus image-only model based on ResNet-50 [31]. The model architecture is illustrated in Fig 2.

2. Fundus image-only model based on EfficientNetV2S [32]. The model architecture is illustrated in Fig 3.

3. Fundus image and patient metadata (age, gender, ethnicity) model. The model architecture is illustrated in Fig 4.

4. Gaussian blur [33] enhanced fundus image model. The model architecture is illustrated in Fig 5.

5. Lightness channel contrast-limited histogram equalization (CLAHE) [34] enhanced fundus image model. The model architecture is illustrated in Fig 6.

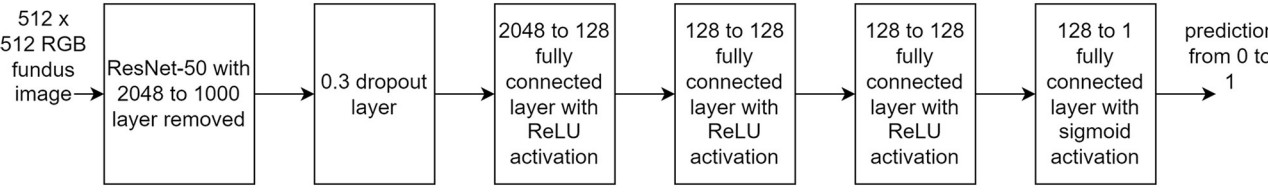

**Fig 2. ResNet block diagram.** Block diagram for baseline ResNet model.

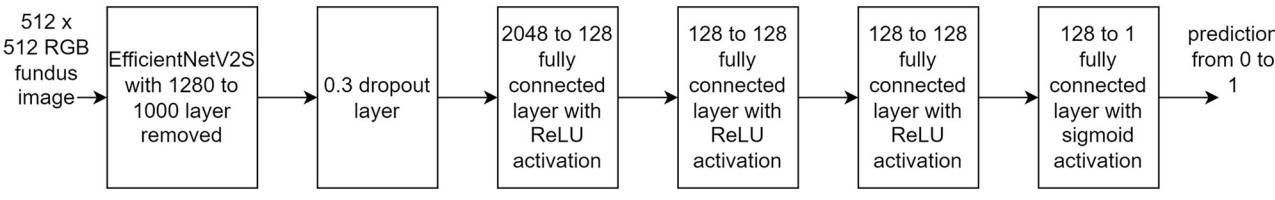

**Fig 3. EfficientNet block diagram.** Block diagram for EfficientNet model.

## Model training and performance evaluation

For all models, input variable normalization was carried out before training. The models were configured as binary classifiers, utilizing the binary cross entropy loss function. A batch size of 32 and the ADAM optimizer with a learning rate of 1e-3 and learning decay of 1e-6 were employed for optimization.

Three different CKD definitions were assigned to individuals in the development and hold-out testing sets. The first definition was based on the single-factor (serum creatinine only) eGFR, the second definition on the two-factor (serum creatinine and cystatin-C) eGFR, and the third definition based on the date chronic renal failure first reported.

For the two eGFR-based definitions, as per prior research [11, 12, 18], individuals with eGFR less than 60 mL/min per 1.73 m$^2$ were classified as CKD-positive and labeled 1. For training, individuals with eGFR greater than 90 mL/min per 1.73 m$^2$ were categorized as CKD negative and labeled 0. Individuals with eGFR between 60–90 mL/min per 1.73 m$^2$ weren't used for training. Internal testing showed that this approach resulted in superior performing models when compared to a singular 60 mL/min per 1.73 m$^2$ threshold. For the clinical definition, individuals with a valid date for chronic renal failure were categorized as CKD positive, and everyone else was classified as CKD negative.

Model performance was then evaluated using eGFR and clinical diagnosis. For evaluation using eGFR, the two model ensembles were evaluated on their ability to distinguish between individuals with eGFR lower than 60 mL/min per 1.73 m$^2$ from all other individuals based on the same eGFR definition they were trained on. For evaluation using clinical diagnosis, the models were evaluated on their ability to distinguish between individuals with reported ICD10 N18 (chronic renal failure) and all other individuals.

The receiver operating characteristics area under the curve (ROC AUC), sensitivity, specificity, and F1 scores were used to measure model performance for all models. For metrics, such as sensitivity, that require a threshold, the optimal threshold was determined using Youden's J statistic [35]. The pairwise t-test was then used to determine the statistical significance between the series metrics generated by the two model ensembles developed under different eGFR definitions. Prior to t-test execution, the Shapiro-Wilk test was used to determine that the differences between the pairs were normally distributed.

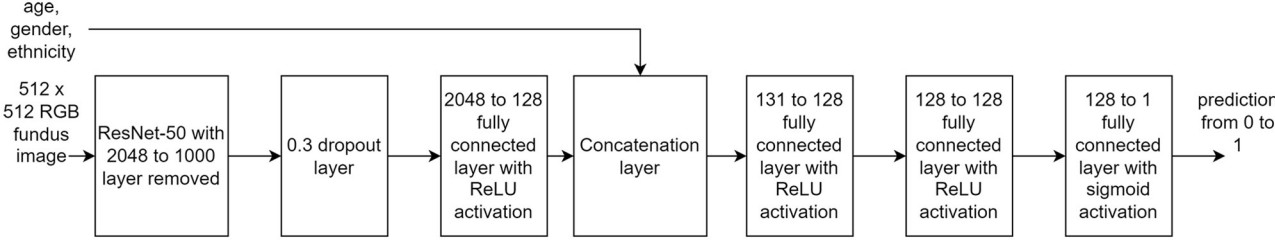

**Fig 4. Metadata model block diagram.** Block diagram for metadata model.

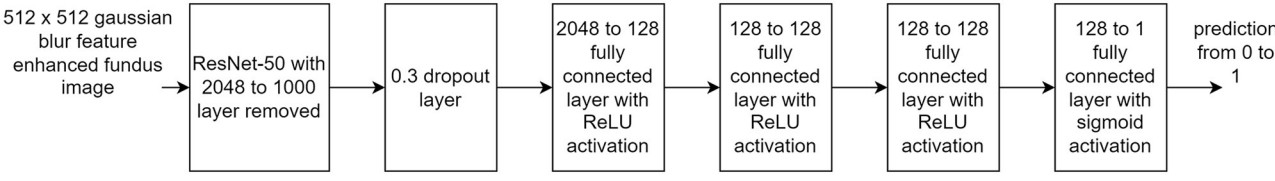

**Fig 5. Gaussian blur enhanced model block diagram.** Block diagram for gaussian blur enhanced model.

## Results

Tables 2 and 3

## Discussion

We found that models trained on labels defined using a two-factor eGFR (serum creatinine and cystatin C) performed better than those trained on a single-factor (serum creatinine only) eGFR when evaluated on the same eGFR that was used for training. Pairwise t-tests in Table 2 showed statistically significant differences in ROC AUC sensitivity specificity and F1 scores across the two model ensembles developed from the respective eGFRs. However, as per Table 3, the differences were no longer statistically significant when the two model ensembles were evaluated on ICD10-based clinical diagnosis. Comparison with existing DL models showed that our best-performing model, the fundus and metadata model developed using the two-factor eGFR, was comparable to the model developed by Kang et al. [12], which reported a sensitivity and specificity of 0.83 and 0.62, respectively. However, our models were inferior to the Chinese CC-FII [18] and Singaporean SEED [11] studies, which reported ROC AUC above 0.9.

An examination of the demographic distributions of the UK Biobank and the datasets used by SEED/CC-FII revealed several key differences that might explain the observed differences in model performance. Firstly, the UK Biobank is predominantly European, whereas the CC-FII/SEED studies were limited to Asian ethnic groups. Furthermore, noticeable disparities in the incidence rates of systemic conditions common to CKD and retinal disorders exist between the UK Biobank and the SEED/CC-FII datasets. For example, the diabetes incidence in the UK Biobank dataset was 5–6%, compared to 28.6% and 29.2% in the Singaporean SEED and Chinese CC-FII studies. For reference, the diabetes incidence rate for people aged 55–65 in China is estimated to be 15.98% [36], indicating that the CC-FII and SEED dataset likely comprise a group of individuals with a higher incidence of diabetes than occurs in the general population. Similarly, there was also a difference in the hypertension incidence rate between the UK Biobank and SEED studies, with rates of 35.2% [37] and 62.5%, respectively.

The proposition that systemic condition incidence is related to the apparent performance of DL classifiers is further supported by inconsistencies in the SEED study's image-only classifier when applied to external test datasets [11]. The authors reported a drop in ROC AUC

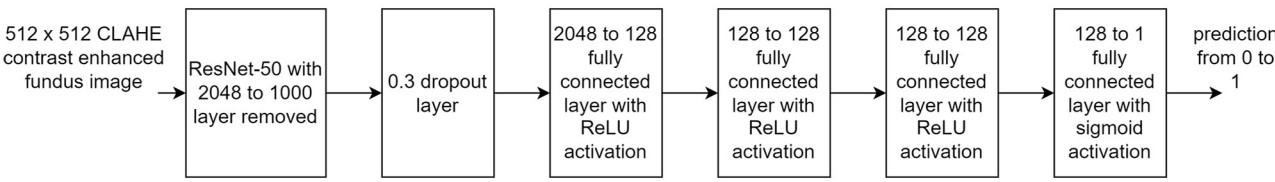

**Fig 6. L-CLAHE enhanced model block diagram.** Block diagram for lightness channel CLAHE enhanced model.

**Table 2. Model performances for evaluation using eGFR.**

| | Models developed using single-factor eGFR (creatinine only) | | | | | Models developed with two-factor eGFR (serum creatinine and cystatin C) | | | | | Pairwise T-test results | | | |
|---|---|---|---|---|---|---|---|---|---|---|---|---|---|---|
| | (1) ResNet | (2) EfficientNet | (3) Metadata | (4) Gaussian blur | (5) CLAHE | (1) ResNet | (2) EfficientNet | (3) Metadata | (4) Gaussian blur | (5) CLAHE | Test statistic | DOF | P value | Mean difference |
| ROC AUC | 0.649 | 0.668 | 0.747 | 0.681 | 0.651 | 0.742 | 0.758 | 0.798 | 0.737 | 0.724 | 8.357 | 4 | P<0.01 (**) | 0.072 |
| Sensitivity | 0.693 | 0.723 | 0.823 | 0.792 | 0.693 | 0.792 | 0.833 | 0.851 | 0.832 | 0.710 | 3.116 | 4 | 0.036 (*) | 0.059 |
| Specificity | 0.543 | 0.55 | 0.582 | 0.496 | 0.552 | 0.597 | 0.584 | 0.533 | 0.628 | 0.663 | 4.288 | 4 | 0.013 (*) | 0.059 |
| F1 score | 0.204 | 0.215 | 0.254 | 0.214 | 0.207 | 0.243 | 0.248 | 0.281 | 0.228 | 0.251 | 6.236 | 4 | P<0.01 (**) | 0.032 |

This table compares model ensembles developed using single-factor vs. two-factor eGFR in classifying groups defined by a 60 mL/min per 1.73 $m^2$ eGFR threshold on the hold-out testing dataset. Models were evaluated on the same eGFR definition that was used for model training.

**Table 3. Model performances for evaluation using reported renal failure.**

| | Models developed using single-factor eGFR (creatinine only) | | | | | Models developed with two-factor eGFR (serum creatinine and cystatin C) | | | | | Pairwise T-test results | | | |
|---|---|---|---|---|---|---|---|---|---|---|---|---|---|---|
| | (1) ResNet | (2) EfficientNet | (3) Metadata | (4) Gaussian blur | (5) CLAHE | (1) ResNet | (2) EfficientNet | (3) Metadata | (4) Gaussian blur | (5) CLAHE | Test statistic | DOF | P value | Mean difference |
| **ROC AUC** | 0.677 | 0.690 | 0.745 | 0.699 | 0.666 | 0.707 | 0.712 | 0.749 | 0.688 | 0.693 | 1.860 | 4 | 0.136 | 0.014 |
| **Sensitivity** | 0.704 | 0.785 | 0.806 | 0.785 | 0.765 | 0.781 | 0.838 | 0.785 | 0.810 | 0.721 | 0.793 | 4 | 0.472 | 0.018 |
| **Specificity** | 0.606 | 0.558 | 0.587 | 0.545 | 0.522 | 0.571 | 0.503 | 0.622 | 0.510 | 0.623 | 0.067 | 4 | 0.950 | 0.002 |
| **F1 score** | 0.244 | 0.247 | 0.265 | 0.241 | 0.228 | 0.251 | 0.240 | 0.275 | 0.235 | 0.256 | 1.008 | 4 | 0.371 | 0.007 |

This table compares model ensembles developed using single-factor eGFR vs. two-factor eGFR in classifying binary groups defined by reported ICD10 N18 (chronic renal failure). Individuals with reported N18 were labeled as 1. All other individuals were labeled as 0.

from 0.911 to 0.733 when the model was used on the external SP2 dataset instead of the internal SEED validation dataset. It was noteworthy that the SP2 dataset comprised younger individuals (mean age 49.9 vs. 58.4 in the training set) who had a lower diabetes incidence rate (9.8% vs. 28.6% in the training dataset) and a lower hypertension rate; (40.6% vs. 62.5%). A similar decrease in performance was also observed on the second BES external validation dataset. A more modest reduction from 0.911 to 0.835 ROC AUC was observed in this case. The review also revealed that the demographics of the BES dataset were more similar to the internal SEED validation set, with older individuals (mean age 64.3) and elevated diabetes (16.8%) and hypertension rates (51%). Signs of this relationship were present in Kang et al.'s study, whereby the authors reported an improvement in model performance (ROC AUC 0.81 to 0.87) by enriching their base dataset with individuals that have poorer glycemic control and higher average HbA1Cs (>10%). In summary, these findings collectively paint a situation where the performance of fundus image-based DL CKD predictors appears to be linked to the incidence of systemic conditions known to be associated with an increased incidence of CKD.

Finally, our results suggest that seeming improvements in metric performance have a limited impact on actual patient diagnosis. Despite statistical differences in metric performance between models developed using a one-factor eGFR and a two-factor eGFR when evaluated on their respective labels, these differences ceased to be statistically significant when the models were assessed on clinically diagnosed CKD as per ICD 10. The behavior implies that despite the apparent performance gains in switching from a one-factor to a two-factor eGFR, the quality and generalizability of the features learned by the models likely remained the same.

We hypothesize that the susceptibility of fundus image-based DL CKD predictors to variations in labeling strategy and systemic disease incidence can be attributed to two inherent limitations in the modeling process. Firstly, eGFR has a P30 value of 67–87% compared to gold standard indices for kidney function [19]. When binary disease state labels are defined using thresholds based on noisy measures, there will be inherent uncertainty in the labels, particularly for individuals close to the decision thresholds. Secondly, despite the widely substantiated pathophysiological connections between the eye and the kidney, including the shared expression of the renin-angiotensin-aldosterone-system (RAAS) [38], structural similarities in basement membranes, and common pathways in oxidative stress, inflammation, and atherosclerosis [5] there is still a paucity of retinal features discernable in color fundus photographs that unequivocally signify CKD while being unrelated to other concomitant systemic conditions, such as hypertension and diabetes. The lowered certainty in ground truth likely makes classifying CKD from fundus photographs inconsistent and dependent on the prevalence of other

systemic conditions in the dataset's population. This contrasts with DL models for diabetic retinopathy, which report consistently good classification performance [39], partly due to features and labels being defined by rigid, visually identifiable physical signatures [8].

In an attempt to account for the inherent noise in a CKD state label defined only by eGFR, we employed the basic strategy of omitting individuals with eGFR values corresponding to stage 1 and stage 2 CKD during model training. However, due to our decision to use an eGFR threshold of 60 mL/min per 1.73 m$^2$ for model evaluation, our overall methodology remains susceptible to this limitation. In clinical practice, the diagnosis of CKD entails a comprehensive assessment that considers eGFR, albuminuria and underlying cause [1, 40]. Neglecting these essential components in favor of a decision threshold purely based on eGFR increases the risk of misclassification. However, to facilitate a meaningful comparison with prior research, we opted to use an eGFR-only method that is consistent with previous studies. Nonetheless, future experiments should look at incorporating albuminuria and supporting clinical data alongside eGFR when defining a label for CKD.

Other factors that affect the pathophysiological connection between the kidney and retina and the robustness of the CKD state labels include the use drugs such as angiotensin converter enzyme (ACE) inhibitors. ACE is a component of the RAAS that regulates both renal and retinal vasculature. As such, the use of ACE inhibitors can impact renal function and retinal appearance. Ideally, the use of ACE inhibitors should then be considered in this study. However, as medication history (field 6177) in UK Biobank is self-reported with only 323 out of 683 individuals with a creatinine only eGFR $<$ 60 mL/min per 1.73 m$^2$ providing information about any medication usage at all, this raised questions about the representativeness of such data. When this shortcoming was evaluated in context of the core aims of this study, we thought it was appropriate to omit this information. Nonetheless, future methods that seek to infer more reliable drug usage statistics from multiple points of reference would be important and could offer valuable insights into the problem. Similarly, despite factors such as the use of steroids, insulin, and dietary habits impacting creatinine levels and eGFR, due to comparable limitations in obtaining this information reliably from the UK Biobank, we did not consider their influence in this study.

The characteristics of the retinal images in the UK Biobank dataset were also a point of interest. The fundus images from the UK Biobank are single field images acquired using non-mydriatic retinal photography. When compared to mydriatic multi-field fundus photographs, as typically used in diabetic retinopathy screening programs, single field images are limited to capturing lesions within a 45-degree view angle. However, to the best of our understanding, there has been no reported correlation between cystatin-C and the propensity for an individual to develop lesions in the peripheral retina. Consequently, using non-mydriatic single field retinal images for model training will likely reduce the sensitivity of all developed models, rather than being biased toward models developed on any specific definition of eGFR. As such, this limitation does not undermine the validity of our study. Further investigation using multi-field, or ultra widefield fundus photographs is required to quantify the possible effects of peripheral lesions on the performance of DL models designed to diagnose CKD.

Finally, it is important to consider the inherent limitations of a DL approach. One such constraint is DL's reliance on large volumes of well-labeled data. This reliance presents both financial and resourcing challenges when attempting to replicate a fundus photograph model intended to classify CKD on new population groups, particularly when non-standard biomarkers such as cystatin-C are used. One future direction to address this limitation could be the application of transfer-learning methodologies. Transfer-learning could provide a pathway to leverage models trained on existing and readily available clinical data (for example hypertension, retinopathy or drusen) to solve the more specialized and data scarce task of classifying

CKD. Furthermore, the black box nature of DL models makes it challenging to conduct a methodical investigation of the mechanisms that underpin the models' outputs. Despite previous studies [11, 12, 15, 18] demonstrating the possibility of utilizing attention maps based methods [41–43] as an approach to explain the workings of their model, this technique is not without issue [44]. Specifically, attention maps highlight a broad fundus area and lack specificity. This makes them open to interpretation bias, especially on a task involving indeterminate features such as the classification of CKD. As such, there is still a need to develop more deterministic methods for DL model explanation that will enable researchers to better understand the impact of systemic conditions and label definitions on the generalization performance of fundus image-based DL CKD predictors.

## Conclusions

Our results show that despite a seeming increase in performance when fundus image-based DL CKD predictors are trained on a two-factor (cystatin C plus serum creatinine) instead of a single-factor (serum creatinine only) eGFR, further evaluation on actual CKD diagnosis showed that differences were no longer significant. We hypothesize that due to a paucity of retinal features uniquely indicative of CKD and a ground truth based on a noisy physiological surrogate of the pathological state, the fundus image-based DL CKD predictor's performance is likely to be influenced by variations in systemic condition incidence, which could vary across different datasets. The development of methods that facilitate a better understanding of DL model behavior, as well the application of a transfer-learning approach to reduce the reliance on scarce clinical data may provide opportunities for further improvement of DL models for classifying CKD from fundus images.

## Author Contributions

**Conceptualization:** Songyang An, David Squirrell.

**Data curation:** Song Yang, Li Xie.

**Formal analysis:** Songyang An, Ehsan Vaghefi, David Squirrell.

**Funding acquisition:** Ehsan Vaghefi.

**Investigation:** Songyang An.

**Methodology:** Songyang An.

**Project administration:** Ehsan Vaghefi, David Squirrell.

**Supervision:** Ehsan Vaghefi, David Squirrell.

**Writing – original draft:** Songyang An.

**Writing – review & editing:** Ehsan Vaghefi, David Squirrell.

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
