## [Decision Letter · Decision Letter 0]

5 Sep 2023

PONE-D-23-21560Examination of alternative eGFR definitions on the performance of deep learning models for detection of chronic kidney disease from fundus photographsPLOS ONE

Dear Dr. An,

Thank you for submitting your manuscript to PLOS ONE. After careful consideration, we feel that it has merit but does not fully meet PLOS ONE’s publication criteria as it currently stands. Therefore, we invite you to submit a revised version of the manuscript that addresses the points raised during the review process.

We look forward to receiving your revised manuscript.

Kind regards,

Ferdinando Carlo Sasso, PhD, MD

Academic Editor

PLOS ONE

Additional Editor Comments:

The authors need to address all issues raised by the two reviewers.

Reviewers' comments:

Reviewer's Responses to Questions

**Comments to the Author**

1. Is the manuscript technically sound, and do the data support the conclusions?

Reviewer #1: Yes

Reviewer #2: Yes

2. Has the statistical analysis been performed appropriately and rigorously? 

Reviewer #1: Yes

Reviewer #2: I Don't Know

3. Have the authors made all data underlying the findings in their manuscript fully available?

Reviewer #1: Yes

Reviewer #2: Yes

4. Is the manuscript presented in an intelligible fashion and written in standard English?

Reviewer #1: Yes

Reviewer #2: Yes

5. Review Comments to the Author

Reviewer #1: I read with great interest the article titled "Examination of alternative eGFR definitions on the performance of deep learning models for detection of chronic kidney disease from fundus photographs" by Songyang An et al.

The paper's design is sound, and the article is logically organized into appropriate sections and subsections. The English is generally well-written, with only minor spelling errors.

Here are the comments and suggested revisions:

1. By evaluating the population according to eGFR levels superior or not to 60 ml/h/mmq, the authors have mixed patients with chronic kidney disease and without it as the first two stages of CKD exist only in case of urinary albumin excretion, which in turn also represent a risk factor for clinical outcomes (doi: 10.3390/diagnostics11020290). Please discuss this in the text and include this in the limitation section.

2. Another limitation is represented by the lack of data about drugs used. Both kidney function and fundus oculi could change according to some drugs usage. It is far well known the effect of angiotensin converter enzymes inhibitors on high blood pressure, as well as on kidney function. In addition, the role of steroids, insulin, protein rich diets could change creatinine levels. Please discuss it and add to limitation section. Furthermore, as age and comorbidities, such as diabetes, appear, the role of multi-drug therapy to control all risk factors is mandatory (doi: 10.1186/s12933-022-01674-7), thus potentially increase the risk of increase of acute kidney injury, rather than chronic.

Reviewer #2: Dear Editor, I’ve read with great interest the draft called “Examination of alternative eGFR definitions on the performance of deep learning models for detection of chronic kidney disease from fundus photographs” by Songyang An et al. However, some issues need to be raised.

- Line 180: The authors write: “for training, individuals with eGFR greater than 90 mL/min per 1.73 m2 were categorized as CKD negative and labeled 0”. What about people with eGFR between 60-90 mL/min per 1.73 m2?

- Nowadays, non-mydriatic fundus photograph is still not the gold-standard method to evaluate diabetic retinopathy. The authors should briefly discuss this point, eventually adding this issue into limitation section.

- I would suggest to briefly discuss, into Discussion section, the pathophysiological elements associated with concomitant kidney and retinal impairment.

- Limitation: DL models and cystatin are expensive methods, which now could not be widely performed. Could the authors briefly discuss those issues into limitation or a perspective and future section?

- Reference 14 needs doi

6. PLOS authors have the option to publish the peer review history of their article (what does this mean?). If published, this will include your full peer review and any attached files.

Reviewer #1: No

Reviewer #2: No

---

## [Author Response · Author response to Decision Letter 0]

12 Oct 2023

A better formatted version of this response is provided in the "Response to reviewers" document that was submitted as a part of this revision.

Editor comment 1

Comment:

Please ensure that your manuscript meets PLOS ONE's style requirements, including those for file naming. The PLOS ONE style templates can be found at https://journals.plos.org/plosone/s/file?id=wjVg/PLOSOne_formatting_sample_main_body.pdf and https://journals.plos.org/plosone/s/file?id=ba62/PLOSOne_formatting_sample_title_authors_affiliations.pdf

Response:

The title authors and affiliations page has been updated as per the template.

The following changes have been made to the main body as per the template:

1. The caption for figure 1 has been updated as per the template, and the image has been uploaded as a .tiff file.

2. Table 1 has been updated as per the template, and a table caption has been added.

3. The caption for figure 2 has been updated as per the template, and the image has been uploaded as a .tiff file.

4. The caption for figure 3 has been updated as per the template, and the image has been uploaded as a .tiff file.

5. The caption for figure 4 has been updated as per the template, and the image has been uploaded as a .tiff file.

6. The caption for figure 5 has been updated as per the template, and the image has been uploaded as a .tiff file.

7. The caption for figure 6 has been updated as per the template, and the image has been uploaded as a .tiff file.

8. Table 2 has been updated as per the template, and a table caption has been added.

9. Table 3 has been updated as per the template, and a table caption has been added.

Editor comment 2

Comment:

Please provide additional details regarding participant consent. In the ethics statement in the Methods and online submission information, please ensure that you have specified (1) whether consent was informed and (2) what type you obtained (for instance, written or verbal, and if verbal, how it was documented and witnessed). If your study included minors, state whether you obtained consent from parents or guardians. If the need for consent was waived by the ethics committee, please include this information.

Response:

This manuscript reports a retrospective study of medical records or archived samples. The previous methods already discussed that all data were fully anonymized before access. The methods section has been updated to mention that UK Biobank waives the requirement for informed consent through its research tissue bank approval. 

Lines 85 to 87:

“This approval waives the requirement for informed consent, which means researchers with applications approved by the UK Biobank do not require separate ethical clearance.”

Editor comment 3:

Comment:

We note that you have indicated that data from this study are available upon request. PLOS only allows data to be available upon request if there are legal or ethical restrictions on sharing data publicly. For more information on unacceptable data access restrictions, please see http://journals.plos.org/plosone/s/data-availability#loc-unacceptable-data-access-restrictions.

Response:

As per the UK Biobank material transfer agreement, there are limitations on data sharing. We have included the details of these restrictions in the updated cover letter.

Reviewer 1 comment 1:

Comment:

By evaluating the population according to eGFR levels superior or not to 60 ml/h/mmq, the authors have mixed patients with chronic kidney disease and without it as the first two stages of CKD exist only in case of urinary albumin excretion, which in turn also represent a risk factor for clinical outcomes (doi: 10.3390/diagnostics11020290). Please discuss this in the text and include this in the limitation section.

Response:

We agree with the reviewer. We’ve amended the manuscript with the following changes:

Lines 289 to 299.

“In an attempt to account for the inherent noise in a CKD state label defined only by eGFR, we employed the basic strategy of omitting individuals with eGFR values corresponding to stage 1 and stage 2 CKD during model training. However, due to our decision to use an eGFR threshold of 60 mL/min per 1.73 m2 for model evaluation, our overall methodology remains susceptible to this limitation. In clinical practice, the diagnosis of CKD entails a comprehensive assessment that considers eGFR, albuminuria and underlying cause [1,40]. Neglecting these essential components in favor of a decision threshold purely based on eGFR increases the risk of misclassification. However, to facilitate a meaningful comparison with prior research, we opted to use an eGFR-only method that is consistent with previous studies. Nonetheless, future experiments should look at incorporating albuminuria and supporting clinical data alongside eGFR when defining a label for CKD.

Reviewer 1 comment 2:

Comment:

Another limitation is represented by the lack of data about drugs used. Both kidney function and fundus oculi could change according to some drugs usage. It is far well known the effect of angiotensin converter enzymes inhibitors on high blood pressure, as well as on kidney function. In addition, the role of steroids, insulin, protein rich diets could change creatinine levels. Please discuss it and add to limitation section. Furthermore, as age and comorbidities, such as diabetes, appear, the role of multi-drug therapy to control all risk factors is mandatory (doi: 10.1186/s12933-022-01674-7), thus potentially increase the risk of increase of acute kidney injury, rather than chronic.

Response:

We agree with this very insightful comment made by the reviewer. We’ve updated the manuscript with the following changes:

Lines 301 to 315.

“Other factors that affect the pathophysiological connection between the kidney and retina and the robustness of the CKD state labels include the use drugs such as angiotensin converter enzyme (ACE) inhibitors. ACE is a component of the RAAS that regulates both renal and retinal vasculature. As such, the use of ACE inhibitors can impact renal function and retinal appearance. Ideally, the use of ACE inhibitors should then be considered in this study. However, as medication history (field 6177) in UK Biobank is self-reported with only 323 out of 683 individuals with a creatinine only eGFR < 60 mL/min per 1.73 m2 providing information about any medication usage at all, this raised questions about the representativeness of such data. When this shortcoming was evaluated in context of the core aims of this study, we thought it was appropriate to omit this information. Nonetheless, future methods that seek to infer more reliable drug usage statistics from multiple points of reference would be important and could offer valuable insights into the problem. Similarly, despite factors such as the use of steroids, insulin, and dietary habits impacting creatinine levels and eGFR, due to comparable limitations in obtaining this information reliably from the UK Biobank, we did not consider their influence in this study.”

Reviewer 2 comment 1:

Comment:

Line 180: The authors write: “for training, individuals with eGFR greater than 90 mL/min per 1.73 m2 were categorized as CKD negative and labeled 0”. What about people with eGFR between 60-90 mL/min per 1.73 m2?

Response:

Thank you for noticing this ambiguity. We’ve updated the manuscript with the following changes:

Lines 189 to 190:

“Individuals with eGFR between 60-90 mL/min per 1.73 m2 weren’t used for training.”

We have also emphasized this point in the discussion on lines 289 to 291.

“In an attempt to account for the inherent noise in a CKD state label defined only by eGFR, we employed the basic strategy of omitting individuals with eGFR values corresponding to stage 1 and stage 2 CKD during model training.”

Reviewer 2 comment 2:

Comment:

Nowadays, non-mydriatic fundus photograph is still not the gold-standard method to evaluate diabetic retinopathy. The authors should briefly discuss this point, eventually adding this issue into limitation section.

Response:

This is an interesting comment made by the reviewer. We have discussed this comment and the implicated limitations on lines 317 to 328:

“The characteristics of the retinal images in the UK Biobank dataset were also a point of interest. The fundus images from the UK Biobank are single field images acquired using non-mydriatic retinal photography. When compared to mydriatic multi-field fundus photographs, as typically used in diabetic retinopathy screening programs, single field images are limited to capturing lesions within a 45-degree view angle. However, to the best of our understanding, there has been no reported correlation between cystatin-C and the propensity for an individual to develop lesions in the peripheral retina. Consequently, using non-mydriatic single field retinal images for model training will likely reduce the sensitivity of all developed models, rather than being biased toward models developed on any specific definition of eGFR. As such, this limitation does not undermine the validity of our study. Further investigation using multi-field, or ultra widefield fundus photographs is required to quantify the possible effects of peripheral lesions on the performance of DL models designed to diagnose CKD.” 

Reviewer 2 comment 3:

Comment:

I would suggest to briefly discuss, into Discussion section, the pathophysiological elements associated with concomitant kidney and retinal impairment.

Response:

We think this is a good suggestion. We have expanded one of our existing discussion points to include a brief discussion about the pathophysiological elements associated with concomitant kidney and retinal impairment.

Lines 276 to 284:

“Secondly, despite the widely substantiated pathophysiological connections between the eye and the kidney, including the shared expression of the renin-angiotensin-aldosterone-system (RAAS)[38], structural similarities in basement membranes, and common pathways in oxidative stress, inflammation, and atherosclerosis [5] there is still a paucity of retinal features discernable in color fundus photographs that unequivocally signify CKD while being unrelated to other concomitant systemic conditions, such as hypertension and diabetes. The lowered certainty in ground truth likely makes classifying CKD from fundus photographs inconsistent and dependent on the prevalence of other systemic conditions in the dataset’s population.”

Reviewer 2 comment 4:

Comment:

Limitation: DL models and cystatin are expensive methods, which now could not be widely performed. Could the authors briefly discuss those issues into limitation or a perspective and future section?

Response:

We think the reviewer raised a fair point. We have included a discussion about this limitation on lines 330 to 338.

“Finally, it is important to consider the inherent limitations of a DL approach. One such constraint is DL’s reliance on large volumes of well-labeled data. This reliance presents both financial and resourcing challenges when attempting to replicate a fundus photograph model intended to classify CKD on new population groups, particularly when non-standard biomarkers such as cystatin-C are used. One future direction to address this limitation could be the application of transfer-learning methodologies. Transfer-learning could provide a pathway to leverage models trained on existing and readily available clinical data (for example hypertension, retinopathy or drusen) to solve the more specialized and data scarce task of classifying CKD.”

Reviewer 2 comment 5:

Comment:

Reference 14 needs doi

Response:

Thank you for noticing this. The reference was originally accessible from Trinity College of Dublin; however, the link is now inactive. We’ve removed this reference from the manuscript and have updated the reference list accordingly.

The only instance where this reference appeared was on line 51:

“research groups have applied DL to either directly diagnose CKD from fundus photographs [11–15]”

Other changes:

Change 1

Change

On lines 129 to 130 the phrase was changed from:

“and date chronic renal failure first reported (ICD10 N18)“

To:

“and date N18 (chronic renal failure) first reported (UK Biobank field 132032) “

Reason

The new wording is more aligned with the field name from the UK Biobank dataset. The UK Biobank field reference has also been included.

Change 2

Change 

In table 1, the row headers for row 3 and row 4 were changed from:

“Individuals with eGFR < 60“and “Corresponding number of images“

To:

“Individuals with eGFR < 60 mL/min/1.73m2 “and “Corresponding number of images 

for Individuals with eGFR < 60 mL/min/1.73m2 “ 

Reason

The original header for row 3 was missing units. The original header for row 4 wasn’t sufficiently descriptive and could be open to interpretation.

Change 3

Change

On lines 199-200 the phrase was changed from:

“between individuals with either current or future reported chronic renal failure”

To:

“between individuals with reported ICD10 N18 (chronic renal failure)

Reason

The new wording is more succinct and better aligned with the wording changes carried out in change 1.

Change 4

Change

Results in tables 2 and 3 were updated.

Reason

While creating the minimal replicable dataset set and then replicating the results, we noticed the following issues:

1. We noticed that the model weight files for the (5) CLAHE weren’t saved properly and were behaving erratically. We decided to retrain the CLAHE models for both single-factor and two-factor eGFRs. However, due to DL’s stochastic nature, there were changes in performance. At most, the fluctuations resulted a change from 0.646 ROC AUC pre-training to 0.666 ROC AUC post training.

2. There were transcription errors in table 3 that caused the specificity values for “models developed with two-factor eGFR” to be shifted one column to the right. For instance, the specificity for model (2) EfficientNet was mistakenly recorded as 0.571, when in fact, this was the specificity for model (1) ResNet. 

None of these changes resulted in an impact on the overall trend or conclusion. Namely, that models trained using a two factor eGFR had a better performance than models trained using a single factor eGFR when evaluated on their respective definitions, but the differences were no longer significant when evaluated on reported renal failure.

Change 5

Change

On lines 242 to 243 the sentence was changed from:

Similarly, the hypertension incidence rate in the UK Biobank was 35.2% [38], whereas, in the SEED and CC-FII studies, the rates were 62.5% and 56%, respectively.

To:

Similarly, there was also a difference in the hypertension incidence rate between the UK Biobank and SEED studies, with rates of 35.2% [37] and 62.5%, respectively.

Reason

The previously stated hypertension incidence rate of 56% for the CC-FII study was a mistake that has now been corrected. All other quoted values have been double-checked for correctness.

Change 6

Change

On lines 357 to 360 the sentence was changed from:

“The development of methods that allow for more deterministic explanation of DL model behavior will be required to facilitate a better understanding of the mechanisms underpinning these models.”

To:

“The development of methods that facilitate a better understanding of DL model behavior, as well the application of a transfer-learning approach to reduce the reliance on scarce clinical data may provide opportunities for further improvement of DL models for classifying CKD from fundus images.”

Reason

The conclusion was updated to reflect the inclusion of additional discussion points that were recommended by the reviewers.

---

## [Decision Letter · Decision Letter 1]

15 Nov 2023

Examination of alternative eGFR definitions on the performance of deep learning models for detection of chronic kidney disease from fundus photographs

PONE-D-23-21560R1

Dear Dr. An,

We’re pleased to inform you that your manuscript has been judged scientifically suitable for publication and will be formally accepted for publication once it meets all outstanding technical requirements.

Kind regards,

Ferdinando Carlo Sasso, PhD, MD

Academic Editor

PLOS ONE

Additional Editor Comments (optional):

Non further comments

Reviewers' comments:

Reviewer's Responses to Questions

**Comments to the Author**

1. If the authors have adequately addressed your comments raised in a previous round of review and you feel that this manuscript is now acceptable for publication, you may indicate that here to bypass the “Comments to the Author” section, enter your conflict of interest statement in the “Confidential to Editor” section, and submit your "Accept" recommendation.

Reviewer #1: All comments have been addressed

Reviewer #2: All comments have been addressed

2. Is the manuscript technically sound, and do the data support the conclusions?

Reviewer #1: Yes

Reviewer #2: Yes

3. Has the statistical analysis been performed appropriately and rigorously? 

Reviewer #1: Yes

Reviewer #2: Yes

4. Have the authors made all data underlying the findings in their manuscript fully available?

Reviewer #1: Yes

Reviewer #2: Yes

5. Is the manuscript presented in an intelligible fashion and written in standard English?

Reviewer #1: Yes

Reviewer #2: Yes

6. Review Comments to the Author

Reviewer #1: The authors appropriately answered to all the issues I raised. in my opinion the paper can be further process for publication

Reviewer #2: It is opinion of this reviewer that the authors have improved the manuscript and all requested comments have been addressed.

7. PLOS authors have the option to publish the peer review history of their article (what does this mean?). If published, this will include your full peer review and any attached files.

Reviewer #1: No

Reviewer #2: No

---

## [Editor Report · Acceptance letter]

20 Nov 2023

PONE-D-23-21560R1 

Examination of alternative eGFR definitions on the performance of deep learning models for detection of chronic kidney disease from fundus photographs 

Dear Dr. An:

I'm pleased to inform you that your manuscript has been deemed suitable for publication in PLOS ONE. Congratulations! Your manuscript is now with our production department. 

Kind regards, 

on behalf of

Professor Ferdinando Carlo Sasso 

Academic Editor

PLOS ONE